Forms of Melanoplus bowditchi (Orthoptera: Acrididae) collected from different host plants are indistinguishable genetically and in aedeagal morphology

Ullah Muhammad Irfan 1 4
Mustafa Fatima 1 5
Kneeland Kate M. 1
Brust Mathew L. 2
Hoback W. Wyatt 3 6 hobackww@unk.edu
Kamble Shripat T. 1
Foster John E. 1
1 Department of Entomology, University of Nebraska , Lincoln, NE , USA
2 Department of Biology, Chadron State College , Chadron, NE , USA
3 Department of Biology, University of Nebraska , Kearney, NE , USA
Huber Dezene
4 Current affiliation: Department of Entomology, University of Sargodha, Pakistan

5 Current affiliation: Department of Entomology, University of Agriculture Faisalabad, Pakistan

6 Current affiliation: Entomology and Plant Pathology Department, Oklahoma State University, Stillwater, OK, USA

Electronic publication date: 2014 Jun 10
Publication date: 2014
Volume: 2
Electronic Location ID: e418
Received 2014 Jan 10; Accepted 2014 May 17
Copyright: © 2014 Ullah et al.
Copyright year: 2014
Copyright holder: Ullah et al.
License: This is an open access article distributed under the terms of the Creative Commons Attribution License, which permits unrestricted use, distribution, and reproduction in any medium, provided the original author and source are credited.
License URL: https://creativecommons.org/licenses/by/3.0/

Keywords: Melanoplus bowditchi, Morphological variation, Ecotype, Deme, Host plant, AFLP

Funding: USDA-APHIS USDA-ARS USDA-APHIS provided funding to support collection of grasshoppers and genetic analysis. USDA-ARS provided funding to support Kate Kneeland. The funders had no role in study design, data collection and analysis, decision to publish, or preparation of the manuscript.

==============================
The sagebrush grasshopper, Melanoplus bowditchi Scudder (Orthoptera: Acrididae), is a phytophilous species that is widely distributed in the western United States on sagebrush species. The geographical distribution of M. bowditchi is very similar to the range of its host plants and its feeding association varies in relation to sagebrush distribution. Melanoplus bowditchi bowditchi Scudder and M. bowditchi canus Hebard were described based on their feeding association with different sagebrush species, sand sagebrush and silver sagebrush, respectively. Recently, M. bowditchi have been observed feeding on other plant species in western Nebraska. We collected adult M. bowditchi feeding on four plant species, sand sagebrush, Artemisia filifolia, big sagebrush, A. tridentata, fringed sagebrush, A. frigidus, and winterfat, Krascheninnikovia lanata. We compared the specimens collected from the four plant species for their morphological and genetic differences. We observed no consistent differences among the aedeagal parameres or basal rings among the grasshoppers collected from different host plants. Amplified Fragment Length Polymorphism markers were used to test the genetic relationships among the grasshoppers. Analysis of Molecular Variance and distance-based Unweighted Pair Group Method with Arithmetic mean dendrogram failed to reveal significant differences. Although the forms showed behavioral and minor color and size differences, the genetic data suggest all forms under study likely interbreed, which indicates they are a single species instead of four species or subspecies. These results indicate that host plant use may influence melanopline phenotype and suggest the need of further genetic analysis of subspecies recognized based on morphology, distribution, and ecology.

Introduction

As a group, grasshoppers are somewhat unusual among herbivorous insects in that most are polyphagous, feeding selectively on plants from a number of unrelated plant families (Otte & Joern, 1977). Because of their polyphagy, most grasshopper species are not expected to experience disruptive selection associated with host choice. There are, however, some grasshopper species with restricted host ranges and a small number that are truly host specific (Otte & Joern, 1977; Sword & Dopman, 1999). Host specific grasshoppers also show differences in development rates, lifespan, and size relating to host use (Traxler & Joern, 1999). In addition, host plant-associated genetic differences have also been observed in the study of Hesperotettix viridis (Thomas) and Schistocera lineata Scudder (Sword, Joern & Senior, 2005).

The sagebrush grasshopper, Melanoplus bowditchi Scudder, was described by Scudder in 1878 (Scudder, 1897). This grasshopper is a phytophilous species that is widely distributed in the grasslands of the western United States. Although it occurs in mixed-grass, shortgrass, desert shrub, and bunchgrass prairies, it feeds almost exclusively on sagebrush species (Mulkern et al., 1969) and its distribution is dependent on sagebrush plants. Six host plants are identified for M. bowditchi in Pfadt (1994), with the primary hosts being silver sagebrush, Artemesia cana, and sand sagebrush, A. filifolia. The other four species of sagebrush, along with silver sagebrush, are found in mixed-grass prairie and are reportedly consumed in minute quantities by M. bowditchi (Pfadt, 1994). The species is potentially damaging, especially for silver sagebrush (Pfadt, 1994). While silver sagebrush is broadly distributed across western North America, sand sagebrush, Artemisia filifolia Torrey, is usually associated with deep sand deposits and serves as the host plant for M. bowditchi in areas where silver sagebrush is limited (Harvey, 1981).

The subspecies, Melanoplus bowditchi bowditchi Scudder was proposed after the description of Melanoplus bowditchi canus Hebard (Hebard, 1925). The original series of M. bowditchi bowditchi was found feeding on A. filifolia, while M. bowditchi canus was collected from big sagebrush, Artemisia tridentata (Hebard, 1925). Melanoplus bowditchi canus is usually dark gray in color and is common in the northern Great Plains. Its preferred food plant is silver sagebrush, although it has also been observed feeding on other sagebrush species. It is normally found on taller plants until after oviposition, when it becomes abundant on shorter plants. It is seldom found on the ground (Anderson & Wright, 1952). In comparison to M. bowditchi canus, M. bowditchi bowditchi has a larger body size, brighter yellow and brown colors, and very clear tegmina (Hebard, 1925).

Hebard (1925) suggested that the gray patterned coloration of M. bowditchi canus was a result of a close relationship to the Melanoplus cinereus group rather than to other forms of the Melanoplus flavidus group (Hebard, 1925). However, geographical differences in host plant use and morphology might also be the result of environmentally triggered variation among populations. For example, for a specialized flea beetle, areas with abundant hosts and frequent oviposition show a high level of host acceptance, resulting in less use of low-ranking hosts (Tahvanainen & Root, 1972). Where preferred plants are uncommon or their availability is obscured by related members of the plant community, thresholds for host acceptance are expected to fall, making the use of other plants more likely (Stanton & Cook, 1983; Wiklund, 1975).

Recent observations have revealed adult M. bowditchi feeding on other Artemisia species and winterfat, Krascheninnikovia lanata. These forms seem to be distinct in both size and color patterns (Fig. 1) and exhibit behavioral differences. Specimens of M. bowditchi collected from fringed sagebrush, Artemisia frigida are exceptionally pallid (Fig. 1) ranging from a pale tan to pale gray and superficially resembling Melanoplus angustipennis (Dodge). In addition, specimens collected from fringed sagebrush are more reluctant to jump than M. bowditchi bowditchi and often must be knocked from the small shrubs to be revealed, similar to reports for M. bowditchi canus. Fringed sagebrush is common in dry, well-drained soils or in disturbed areas. In mixed-grass prairie it is found with western wheatgrass Pascopyrum smithii, blue grama Bouteloua gracilis, and winterfat.

Figure 1 Lateral view of Melanoplus bowditchii grasshoppers collected feeding on (A) sand sagebrush, (B) fringed sagebrush, (C) winterfat, and (D) big sagebrush.

Specimens of M. bowditchi that are collected feeding on winterfat are silvery gray. Behaviorally, specimens of this form are much more wary than those collected from fringed sagebrush and silver sagebrush and jump readily like M. bowditchi bowditchi.

The objective of this research was to examine the genetic and aedeagal characteristics for the adults of M. bowditchi associated with different sagebrush species and to test our hypotheses that the forms associated with host plants are four distinct species.

Materials & Methods

Insects

A series of adult M. bowditchi were collected from four host plants, sand sagebrush, A. filifolia, big sagebrush, A. tridentata, fringed sagebrush, A. frigidus, and winterfat, Krascheninnikovia lanata (Table 1). We also collected a distant outgroup, the mottled sand grasshopper, Spharagemon collare (Scudder) from bare soil patches in western Nebraska. Specimens were identified based on the available literature of Bruner (1897), Scudder (1897), Pfadt (2002) and Brust, Hoback & Wright (2008).

Table 1 State, county location, host plant and date of collection for Melanoplus bowditchii specimens used in aedeagus analysis.

State	County	Location	Host plant	Date	Quantity	
Nebraska	Dawes	8 km S of Chadron	Fringed sagebrush Artemisia frigida	July 24, 2010	3	
South Dakota	Fall River	24 km N of Ardmore	Big sagebrush Artemisia tridentata	August 21, 2010	3	
Nebraska	Morill	14.4 km SW of Alliance	Sand sagebrush Artemisia filifolia	July 24, 2010	3	
Nebraska	Scotts Bluff	12 km N of Minatare	Winterfat Krascheninnikovia lanata	July 17, 2010	4	

Aedeagal analysis

For aedeagal studies, abdomens from three or four grasshoppers collected from each plant species were examined (Table 1). In each case, the terminal part of the abdomen was separated, intestinal contents removed, and the remaining structure soaked in a solution of 5% NaOH for 8–10 h, transferred to 70% ethanol for 10 min, and the aedeagi removed. Aedeagi were cleaned under a dissecting microscope to remove connective tissue. They were preserved in 70% ethanol until examination. Photographs of aedeagi were taken through a dissecting microscope. A visual comparison was made of the structure of the terminal end of the aedeagus, and of the paramere structures.

Genetic analysis—extraction and quantification of DNA

The locations, date of collection, number of specimens and plants from which specimens were collected for genetic analysis are presented in Table 2. Hind femora of specimens from each host plant were preserved in 95% ethanol and stored at −80 °C prior to genetic studies. A total of 35 grasshoppers were examined for genetic differences (Table 2). DNA was isolated from the hind femur of each form specimen using acetyletrimethylammonium bromide (CTAB) extraction protocol modified from Doyle & Doyle (1987). Each hind femur was placed in an autoclaved 1.5 ml microcentrifuge tube and rinsed for 10 min in Nanopure® water. The entire hind femur was homogenized in 250 µl CTAB buffer (100 mM Tris–HCl, 1.4 M NaCl, 0.02 M EDTA, 2% CTAB, and 0.2% β-mercaptoethanol) using sterile white quartz sand and plastic pestles. Another 250 µl of CTAB was added to the tubes to make a volume of 500 µl. RNase A (15 µl of 0.05 g ml−1) was added to each tube, and incubated for 2 h at 65 °C. Proteinase K (15 µl of 0.02 g ml−1) was added, and incubated for 1 h at 37 °C. Samples were centrifuged for 5 min at 20 °C and 12,000 rpm. The supernatant from each tube was transferred to new autoclaved tubes and the tissue discarded. Chloroform: isoamyl alcohol (24:1) (500 µl) was added to the supernatant, and tubes were centrifuged at room temperature for 20 min at 12,000 rpm. The upper aqueous layer was transferred to new 1.5 ml Eppendorf tubes and the chloroform: isoamyl alcohol step was repeated to isolate the refined top aqueous phase. Chilled isopropanol (400 µl, −20 °C) was added to the tubes to precipitate the DNA, and samples were stored overnight at 4 °C.

Table 2 Collection information for specimens of M. bowditchi from different host plants and outgroup, Spharagemon collare used in genetic analysis.

Specimens of S. collare were collected with sweep nets from bare soil.

Species	State	County	Location	Host plant	Date	Quantity	
Melanoplus bowditchi	Nebraska	Dawes	8 km S of Chadron	Fringed sagebrush
Artemisia frigida	July 24, 2010;
August 7, 2010	8	
Melanoplus bowditchi	South Dakota	Fall River	24 km N of Ardmore	Big sagebrush
Artemisia tridentata	August 21, 2010	3	
Melanoplus bowditchi	Nebraska	Morrill	17.7 km SW of Alliance,
6.4 km E of Broadwater	Sand sagebrush
Artemisia filifolia	July 9, 2010	11	
Melanoplus bowditchi	Nebraska	Scotts Bluff	12 km N of Minatare	Winterfat
Krascheninnikovia lanata	July 18, 2010	10	
Spharagemon collare (outgroup)	Nebraska	Dawes	4.8 km S of Chadron	None	August 22, 2010	3	

Samples were centrifuged at 12,000 rpm for 30 min at 4 °C, to form a pellet of DNA at the bottom of the tube. The supernatant was discarded, and the DNA was washed with 400 µl of chilled absolute ethanol followed by centrifugation for 5 min. The supernatant was decanted and the wash was repeated using 70% ethanol. Tubes were centrifuged again for 5 min, then the ethanol was removed and the samples allowed to air dry. The pelleted DNA was suspended in 50 µl autoclaved 1× TE buffer (10 mM Tris–HCL, 0.1 mM EDTA).

AFLP-PCR methods for genetic analysis

The Amplified Fragment Length Polymorphism (AFLP) technique, modified from Vos et al. (1995) was used for DNA analysis. AFLP consisted of digestion using MseI and EcoRI restriction enzymes, ligation of specific nucleotide adapters, a preselective amplification using universal primers, and a selective amplification using specific primer pairs.

Template preparation

Restriction digestion was performed using 1.25 µl NEB Buffer 4 (New England Biolabs, Foster City, CA), 0.125 µl bovine serum albumin (New England Biolabs), 0.0625 µl EcoRI, 0.0625 µl MseI (New England Biolabs), 3.94 µl Nanopure® water and 7 µl of ∼20 ng/µl DNA template for a total volume of 12.5 µl. The restriction digestion was incubated on a GeneAmp 2720 thermal cycler (Applied Biosystems, Foster City, CA, USA) at 37 °C for 2.5 h. A ligation mixture (5 µl) consisting of 0.5 µl EcoRI and MseI prepared adapters, (Integrated DNA Technologies, Coralville, IA, USA), 0.5 µl T4 DNA ligase, 0.15 µl 10× T4 DNA ligase buffer (New England Biolabs), and 3.35 µl Nanopure® water was dispensed into the tubes containing the digestion product and incubated at 25 °C for 8 h. The ligation product was then diluted using 135 µl of 1× TE buffer. A Nanodrop® spectrophotometer (Thermo Fisher Scientific, Walltham, MA, USA) was used to determine the quantity and quality of DNA in ng/µl from each tube.

Preamplification

A preamplification mix consisting of 10 µl Preamlification Primer Mix II (LI-COR Biosciences, Lincoln NE, USA), 0.25 µl Amplitaq 360 DNA polymerase, 0.75 µl 25 mM MgCl2, and 1.25 µl 10× PCR buffer II (Applied Biosystems, Foster City, CA) was mixed with 1.25 µl of ligation product and run on a PCR program of 20 cycles (30 s at 94 °C, 1 min at 56 °C, and 1 min at 72 °C), then stored at 4 °C. Nanopure® water was used to dilute the product to a ratio of 1:20. Nucleotide sequences of adapters, preamplification primers and selective primers tested are shown in Table 3. A combination of different primer sets was tested and the best working primer sets for grasshopper DNA were chosen (Table 4).

Table 3 Nucleotide sequences of adapters, preamplification primers and selective primers used in this study.

Sequences were described by Vos et al. (1995).

Oligonucleotide	Purpose	Sequence	
EcoRI-1 (forward)	Adapter	5′-CTCGTAGACTGCGTACC-3′	
EcoRI-2 (reverse)	Adapter	5′-AATTGGTACGCAGTCTAC-3′	
MseI-1 (forward)	Adapter	5′-GACGATGAGTCCTGAG-3′	
MseI-2 (reverse)	Adapter	5′-TACTCAGGACTCAT-3′	
E (N + 0)	Preamplification primer	5′-GACTGCGTACCAATTC-3′	
M (N + 1)	Preamplification primer	5′-GATGAGTCCTGAGTAAC-3′	
M-CAA	Selective primer	5′-GATGAGTCCTGAGTAACAA-3′	
M-CTC	Selective primer	5′-GATGAGTCCTGAGTAACTC-3′	
M-CAG	Selective primer	5′-GATGAGTCCTGAGTAACAG-3′	
E-AAC	Selective primer	5′-GACTGCGTACCAATTCAAC-3′	
E-ACT	Selective primer	5′-GACTGCGTACCAATTCACT-3′	
E-AGG	Selective primer	5′-GACTGCGTACCAATTCAGG-3′	
E-ACA	Selective primer	5′-GACTGCGTACCAATTCACA-3′	

Table 4 Selective primer combinations used for AFLP analysis and number of marker bands obtained for each of six types of four-base pair primer sets.

Primer set	EcoRI	MseI	Number of markers	
1	CAAC	ACAA	93	
2	CAAC	ACAG	112	
3	CAAC	ACTC	54	
4	CACA	ACAG	41	
5	CACT	ACAG	86	
6	CAGG	ACTC	83	

Selective amplification

The selective PCR mix was prepared consisting of 1.2 µl 10× PCR buffer II, 0.72 µl 25 mM MgCl2, 0.24 µl(10 mM) deoxynucleotide triphosphate mix, 0.07 µl Amplitaq 360 DNA polymerase (Applied Biosystems), 0.5 µl of Msel primer (5.0 µM), 0.3 µl EcoRI (1.0 µM) IRD-700 labeled primer (Integrated DNA Technologies, Coralville, IA), 6.97 µl nanopure® water, and 1.5 µl of the preamplification template DNA. This step was performed in the dark due to light sensitivity of the labeled primers. Selective amplification was performed on a GeneAmp 2720 thermal cycler (Applied Biosystems) with one pre-PCR cycle (30 s at 94 °C, 30 s at 65 °C, 1 min at 72 °C), 12 cycles of 30 s at 94 °C, 30 s at 65 °C → 56 °C, 60 s at 72 °C, and 23 cycles of 30 s at 94 °C, 30 s at 65 °C → 56 °C and 60 s at 72 °C. Blue stop solution (LI-COR Biosciences, Lincoln, NE) (2.5 µl) was used to end the reaction. The product was then denatured for 3 min at 94 °C and stored at −20 °C.

Data scoring and analysis

The amplified DNA was electrophoresed in KBplus 6.5% polyacrylamide gel on a GeneReader 4200 DNA analyzer (LI-COR Biosciences) which detects bands through infrared inflorescence. An IRDye-700 labeled 50–700 bp size standard was used to estimate fragment size. The correlation of % coefficient of variation and the total number of markers was estimated using Bootsie (J Payne, E Lindroth, KM Kneeland, SR Skoda, JE Foster, unpublished data, 2011).

Gels were scored on Saga Generation 2 version 3.3.0 software (LI-COR Biosciences). Data were converted to a binary matrix for analysis, with 1 = presence of a band and 0 = absence of band. Data were analyzed using Arlequin version 3.5 (Excoffier & Lischer, 2010) and Popgene version 1.32 (Yeh, Yang & Boyle, 1999). Phylogenetic relationships were examined using distance-based methods for the different host associated M. bowditchi. An Unweighted Pair Group Method with Arithmetic mean (UPGMA) dendrogram was constructed using presence/absence characters with the software package PAUP* version 4.0beta (Swofford, 2001). Bootstrapping was performed with 1,000 replicates.

Figure 2 Dorsal view of the aedeagus of Melanoplus bowditchi collected from (A) sand sagebrush, (B) fringed sagebrush, (C) winterfat, and (D) big sagebrush.

Results

Aedeagus examination

All examined grasshoppers collected from different host plants had similar aedeagi. The structure and angle of parameres were similar among specimens (Figs. 2 and 3). The mean (±1 S.E.) paramere lengths (0.81 ± 0.07 mm) were shortest in grasshoppers collected from fringed sagebrush and longest in grasshoppers collected from winterfat (0.98 ± 0.02 mm); however, differences were not significant (ANOVA, P = 0.085). Aedeagal lengths were also not significantly different, although specimens collected from sand sagebrush had a mean length of 0.81 + 0.02 mm compared to those from winterfat, which had a mean length of 0.63 ± 0.02 mm (ANOVA, P = 0.054).

Figure 3 Lateral view of the aedeagus of Melanoplus bowditchi collected from (A) sand sagebrush, (B) fringed sagebrush, (C) winterfat, and (D) big sagebrush.

Genetic variation study

The M. bowditchi from different host plants were initially screened for a total of 10 primer pairs of which six primer pairs (Table 4) were used for analysis. A total of 469 markers were scored using the six primer pairs and 63% of the loci were polymorphic. Using Bootsie examination (J Payne, E Lindroth, KM Kneeland, SR Skoda, JE Foster, unpublished data, 2011), approximately 96% of the variation in the M. bowditchi populations was captured with the chosen markers.

A dendrogram was constructed using a distance-based Unweighted Pair Group Method with Arithmetic mean (UPGMA). The UPGMA analysis did not reveal significant genetic structure differences among the M. bowditchi collected from different host plants and there were few nodes with bootstrap values greater than 70% (Fig. 4). The molecular sequences of grasshoppers collected from winterfat were spread across the dendrogram (Fig. 4).

Figure 4 Distance-based Unweighted Pair Group Method with Arithmetic mean (UPGMA) dendrogram of M. bowditchi grasshoppers using 1,000 bootstrap replicates.

The dendrogram shows the relationship among individuals. Numbers indicate bootstrap support >50% for populations collected from different host plants.

The M. bowditchi populations were arranged in two groups based on host plant associations and descriptions of recognized subspecies (Hebard, 1925; Pfadt, 1994). Grasshoppers collected from winterfat were paired with those collected from fringed sagebrush (the Melanoplus bowditchii canus group) and those collected from fringed sagebrush were paired with those collected from sand sagebrush (the Melanoplus bowditchi bowditchi group). The AMOVA showed the majority of molecular variation (86.8%) occurred within populations (Table 5). Only 7.9% of the genetic variation occurred among populations within groups while the remaining 5.3% was due to the variation among groups (Table 6). Nei’s (1987) gene diversity (GST) is described as the coefficient of gene differentiation, while fixation index (FST) is the measure of differentiation in sub-populations and is only applicable when there are only 2 alleles at a locus. Nei’s genetic diversity (GST) is analogous to Wright’s genetic divergence (FST). GST measures the degree of differentiation in multiple populations. The genetic divergence (FST) and gene diversity (GST) were low (0.1320 and 0.0879 respectively) while the Nm values (5.1905) were high (Table 6), indicating extensive gene flow among populations.

Table 5 Analysis of Molecular Variance (AMOVA) results and fixation indices.

Significance was tested with 1,023 permutations. Group 1 was collected from sand sagebrush and fringed sagebrush and Group 2 was collected from winterfat and big sagebrush.

Source of variation	d.f.	Sum of
squares	Variance
components	Percentage of
variation	
Among groups	1	200.253	4.69095 Va	5.30	
Among populations
within groups	2	257.326	6.99008 Vb	7.90	
Within populations	28	2150.327	76.79740 Vc	86.80	
Total	31	2607.906	88.47843		
Fixation indices	
FST: 0.13202	

Table 6 Analysis of Nei’s genetic diversity in subdivided populations.

Low GST values suggest diversity among populations, and very high Nm values (>1.0) indicate significant gene flow between grasshopper populations. Group 1 was collected from sand sagebrush and fringed sagebrush and Group 2 was collected from winterfat and big sagebrush.

	Ht	Hs	GST	Nm	
Group 1	0.2843	0.2266	0.2030	1.9630	
Group 2	0.2862	0.2665	0.0690	6.7499	
All populations	0.3127	0.2853	0.0879	5.1905	
Notes.

Ht Total diversity

Hs Diversity within populations

GST Diversity among populations

Nm Estimate of gene flow based on GST

Discussion

A number of phytophagous insect species contain locally adapted host specific populations, although they utilize a number of host plants across their range (Futuyma & Peterson, 1985; Thompson, 1994). This phenomenon is only occasionally reported for grasshoppers (Sword & Dopman, 1999). The sagebrush grasshopper, M. bowditchi, feeds on several species of sagebrush, although A. cana and A. filifolia serve as the main host plants (Mulkern et al., 1969; Pfadt, 1994). The M. bowditchii subspecies differ somewhat in distribution, with M. bowditchi bowditchi found in the southern grass plains and M. bowditchi canus found in the northern sagebrush plains, while the ranges broadly overlap in Wyoming and southwestern South Dakota (Hebard, 1929). The geographical distribution of the host plants is very similar to the range of the subspecies of M. bowditchi (Pfadt, 1994), and the feeding preference of this grasshopper has been shown to vary based on local plant availability. Examination of the crop content of M. bowditchi collected from North Dakota showed feeding on silver sagebrush, while the populations from western Nebraska ate sand sagebrush (Pfadt, 1994). Even though there are some differences in distribution, color, size, and host use between the two described subspecies of M. bowditchi, it is important to question the rationale of naming a subspecies solely on these characteristics. Further, it is important to test if M. bowditchi feeding on other host plants represent cryptic species or additional subspecies.

In Nebraska and South Dakota, we identified four populations of M. bowditchi feeding on different plant species where other host plants were not present. Specimens exhibited behavioral differences and slight color and size differences consistent with previous descriptions of subspecies (Fig. 1). Grasshoppers collected from sand sagebrush and winterfat were both very active, either jumping and flying to another host plant when approached within two meters or actively moving into the basal branches. In contrast, specimens collected from fringed sagebrush generally did not fly far and had to be disturbed before they jumped or flew. Individuals found feeding on big sagebrush were more sedentary than most other Melanoplus forms (M Brust, pers. obs., 2010). In our examination of male genitalia, we did not find consistent differences. We recognize that we had a small sample size and that a series of more individuals from each host would be better. However, among our samples, the aedeagi appeared similar and there were no significant differences in overall length or paramere angle or lengths (Figs. 2 and 3) suggesting that they are physically able to interbreed. Even if morphological differences had been noted, breeding tests would be required as variation in aedeagi occurs among individuals of a species and support for the hypothesis of genitalic incompatibility among species is controversial (Eberhard et al., 1998; Masly, 2012). Thus, genetic testing of gene flow can aid in determining the presence of populations and incipient species.

We found no genetic differentiation or distinct lineages for M. bowditchi in relation to different host plants (Fig. 4) despite collecting grasshoppers from populations located more than 230 km apart (Ardmore, SD and Scottsbluff, NE). We found GST values between 0.06 and 0.2. A GST value of 1 would indicate nearly complete isolation of subpopulations while 0 indicates no isolation. A GST value greater than 0.5 indicates some genetic isolation among subpopulations (Nei, 1987). Thus, the low GST in this study reflects the relative measure of variation among subpopulations with reference to total variation (Table 6). In this study, we were unable to identify any clusters in the dendrogram (Fig. 4) that could separate the populations of M. bowditchi into distinctive groups.

These results are similar to the conclusions of Brust et al. (2010) who found no genetic differences among M. foedus foedus (Scudder), M. foedus fluviatilis Bruner, and Melanoplus packardii Scudder. Also, Chapco & Litzenberger (2002), found no genetic differences between M. foedus and M. packardii nor between Melanoplus angustipennis (Dodge) and Melanoplus femurrubrum (De Geer). The Analysis of Molecular Variance indicates that most of the variation (86.8%) was within populations with a small portion (5.3%) observed among groups, suggesting frequent interbreeding. Similarly, the variation among M. bowditchi from different host plants was low, supporting consistent gene flow. The FST value of 0.1320 supports the conclusion that genetic exchange occurring among the four subpopulations was sufficient to prevent either genetic differentiation or structuring into genetically differentiated subpopulations of M. bowditchi.

There is support in the literature for host plants resulting in distinct phenotypes. For example, Miller (1987) and Futuyma (1990) documented host-specific phenotypes in papilionid butterflies and Ophraella leaf beetles. The grasshopper specimens in this study were collected as adults feeding on specific plants, and they differed in color and behavior; however, they do not appear to have consistent genetic differences.

The use of the trinomial for M. bowditchi appears invalid; however, we found color morphs that differed in behavior associated with a different host and geography. These differences in phenotype appear to be influenced by the environment. These differences are potentially related to the diet, but other environmental factors may play a role. It is further unknown how variable this species is west of the Rocky Mountains. Thus, further detailed investigations for M. bowditchi with morphological and behavioral differences associated with host-plant use should be conducted and the genetic variation among forms should be investigated on a larger scale.

We are thankful to Kerri-Farnsworth Hoback for helpful comments on an earlier version of this manuscript.

Additional Information and Declarations

Competing Interests

Author Contributions

The authors declare there are no competing interests.

Muhammad Irfan Ullah conceived and designed the experiments, performed the experiments, analyzed the data, wrote the paper, prepared figures and/or tables.

Fatima Mustafa conceived and designed the experiments, performed the experiments, analyzed the data, wrote the paper.

Kate M. Kneeland conceived and designed the experiments, performed the experiments, analyzed the data.

Mathew L. Brust conceived and designed the experiments, prepared figures and/or tables, reviewed drafts of the paper.

W. Wyatt Hoback conceived and designed the experiments, analyzed the data.

Shripat T. Kamble conceived and designed the experiments, editing and interpretation of the manuscript.

John E. Foster conceived and designed the experiments, analyzed the data, contributed reagents/materials/analysis tools.

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
