# Peer review of "Forms of Melanoplus bowditchi (Orthoptera: Acrididae) collected from different host plants are indistinguishable genetically and in aedeagal morphology"

_PeerJ, doi:10.7717/peerj.418_

## Round 0.1 · original submission · Minor Revisions

· Academic Editor

Minor Revisions

Both reviewers suggest accepting this MS with minor revisions. Both have made a few comments separate from the MS, and have also annotated the MS with a number of excellent comments, suggestions, and questions. Please carefully review the annotated PDFs.

The authors should review and respond to all of the reviewers' comments (etc.) as doing so will help to further improve this MS.

One reviewer wrote this:

"The total number of grasshoppers samples collected (13) was too small for either the aedeagus comparison or DNA extractions. Are these grasshoppers too rare to collect? I would have liked to see more samples from each collection site/host plant to make sure representative samples of each type were collected."

It is important that the authors respond specifically to this question.

I also noticed in the PDF form of the MS that the color plates (Figures 1, 2, and 3) are somewhat pixelated and not completely clear. The figures provided separately from the MS were somewhat better. The authors should ensure the high quality of the figures in the final MS.

Both reviewers have indicated that they are fine with their reviews being made public. In PeerJ, it is always up to the author to make a final decision on making reviews public. I would strongly encourage the authors to do so, as it adds considerable value to the final output.

Thank you for your submission to PeerJ.

·

Basic reporting

Thank you for the opportunity to review this article. I have made my edits on the paper itself (attached). The following is additional feedback. Overall, this is a very interesting research article. The authors have raised an important and legitimate question related to the grouping of the supspecies/types of this particular grasshopper. Because this grouping were done quite a long time ago, it is important to reexamine the validity of the grouping. I am glad to see that they have used some of the modern tools that are available at their disposal such as DNA testing to validate their work. This a valuable project and would recommend for this to be accepted for publication in PeerJ.
Here is a couple of feedback for editor to consider:
1. In a couple of places in the article, the authors mention that these grasshopper types can interbreed. I have suggested making changes so that it would read something like this “most likely this is the same species and not different types, which implies that they may interbreed. Since they have not actually tested if these types interbreed, it is not relevant to mention definitively that these types can interbreed. Actually, this article will be much stronger if they have done a lab trial to determine if they actually interbred. Perhaps a follow up work to this paper…
2. The total number of grasshoppers samples collected (13) was too small for either the aedeagus comparison or DNA extractions. Are these grasshoppers too rare to collect? I would have liked to see more samples from each collection site/host plant to make sure representative samples of each type were collected.

Experimental design

See notes above...

Validity of the findings

See notes above...

Additional comments

See notes above...

·

Basic reporting

Generally, all PeerJ reporting policies appear to have been followed. There are some areas within the manuscript that, on minor rewriting, will be more clear and concise (these 'edits' were placed on the pdf file which is attached elsewhere in the review).

Experimental design

The manuscript meets the requirements expressed in the Scope of the Journal. The submission clearly defined the question and was conducted rigorously. Methods were adequately described to be reproducible. Some comments, intended to ehance clarity, were placed on the pdf file.

Validity of the findings

Data were robust as were conclusions.

Additional comments

The manuscript is, generally, well written. Several comments were placed on the pdf file; these are intended to point out some oversights, editing issues and to improve clarity. This reviewer recommends acceptance after revisions.

---

## Round 0.2 · accepted · Accept

· Academic Editor

Accept

I have read the rebuttal and reviewed the revised MS and find the revisions to be adequate.

Please consider making the review documents public, as it is a good record of the process of publication of these results.

Thank you for supporting PeerJ by submitting your research results for review.